# Cadherin Expression Profiles Define Glioblastoma Differentiation and Patient Prognosis

**DOI:** 10.3390/cancers16132298

**Published:** 2024-06-22

**Authors:** Carolina Noronha, Ana Sofia Ribeiro, Rita Carvalho, Nuno Mendes, Joaquim Reis, Claudia C. Faria, Ricardo Taipa, Joana Paredes

**Affiliations:** 1Neurosurgery Department, Hospital de Santo António, Centro Hospitalar e Universitário do Porto, 4050-366 Porto, Portugal; cnoronha@ipatimup.pt (C.N.); jlreis@icbas.up.pt (J.R.); 2Cancer Metastasis, i3S, Institute for Research and Innovation in Health, University of Porto, 4200-135 Porto, Portugal; aribeiro@ipatimup.pt (A.S.R.);; 3FMUP—Faculty of Medicine, University of Porto, 4200-135 Porto, Portugal; 4IPATIMUP—Institute of Molecular Pathology and Immunology, University of Porto, 4200-135 Porto, Portugal; nmendes@ipatimup.pt; 5Histology and Electron Microscopy, i3S, Institute for Research and Innovation in Health, University of Porto, 4200-135 Porto, Portugal; 6Neurosurgery Department, Hospital de Santa Maria, Centro Hospitalar Universitário Lisboa Norte, 1649-028 Lisbon, Portugal; claudiafaria@medicina.ulisboa.pt; 7IMM—Instituto de Medicina Molecular João Lobo Antunes, Faculty of Medicine, University of Lisbon, 1649-028 Lisbon, Portugal; 8Neuropathology Department, Hospital de Santo António, Centro Hospitalar Universitário de Santo António, 4050-342 Porto, Portugal; rtaipa.neuropat@chporto.min-saude.pt; 9UMIB—Unit for Multidisciplinary Research in Biomedicine, ICBAS—School of Medicine and Biomedical Sciences, University of Porto, 4050-346 Porto, Portugal; 10ITR—Laboratory for Integrative and Translational Research in Population Health, 4050-600 Porto, Portugal

**Keywords:** glioblastoma, N-cadherin, E-cadherin, P-cadherin, EMT

## Abstract

**Simple Summary:**

Epithelial to mesenchymal transition (EMT) programme is central to various cancers, however how this programme applies to glioblastoma, an aggressive primary brain tumor, remains unknown. In particular, the cadherin switch which involves E-cadherin down-regulation and N-cadherin upregulation, is considered a marker of the EMT in epithelial cancers. Given the knowledge gap in the EMT and cadherins expression in GBM, we studied these proteins (E-, P- and N-cadherin) expression in a large cohort of GBM, extensively characterized with clinical, imaging, neuropathological, treatment and survival data. Our results propose that cadherin expression subgroups reflect an EMT-like programme in GBM and predict patient prognosis.

**Abstract:**

Cadherins are cell–cell adhesion proteins which have been strongly implicated in cancer invasion, dissemination and metastasis capacity; thus, they are key players in the epithelial-to-mesenchymal transition (EMT) program. However, their role in glioblastoma (GBM), a primary central nervous system aggressive tumor, remains to be clarified. N-, E- and P-cadherin expression was analyzed on a large series of GBMs, characterized with clinical, imaging and neuropathological parameters, as well as with patients’ survival data. In addition, cadherins’ expression was studied in match-recurrent cases. Using TCGA data, cadherin expression profiles were also evaluated according to GBM transcription subtypes. N-cadherin expression was observed in 81.5% of GBM, followed by E-cadherin in 31% and P-cadherin in 20.8%. Upon tumor recurrence, P-cadherin was the only significantly upregulated cadherin compared with the primary tumor, being positive in 65.8% of the cases. Actually, P-cadherin gain was observed in 51.4% of matched primary-recurrent cases. Cadherins’ co-expression was also explored. Interestingly, E- and N-cadherin co-expression identified a GBM subgroup with frequent epithelial differentiation and a significant survival benefit. On the other hand, subgroups with P-cadherin expression carried the worse prognosis. P- and N-cadherin co-expression correlated with the presence of a mesenchymal phenotype. Expressions of isolated P-cadherin or E- and P-cadherin co-expression were associated with imaging characteristics of aggressiveness, to highly heterogeneous tumors, an d to worse patient survival. Classical cadherins co-expression subgroups present consistent clinical, imaging, neuropathological and survival differences, which probably reflect different states of an EMT-like program in GBM.

## 1. Introduction

Cadherins are membranous calcium-dependent proteins crucial for cell–cell adhesion [1] and thus are key players in a variety of cellular processes [2,3]. Most frequently, cadherins assemble into adhesive junctions between cells expressing the same cadherin subtype; however, their expression can also be highly heterogeneous, since cells can express more than one cadherin subtype [4,5]. Importantly, through their intracellular cytoplasmic tail, cadherins associate with numerous signaling proteins, participating in a dynamic process dependent on the cellular context and which culminates in the regulation of physiologic mechanisms, such as cell proliferation, differentiation, and migration [3,6,7,8,9]. Most likely, qualitative and quantitative differences in the expression of different cadherin subtypes confer tissue-specific characteristics and functional roles [2,10].

Epithelial (E-), neuronal (N-) and placental (P-) cadherins are classical cadherins, the first members of the superfamily to be identified (*CDH1*, *CDH2*, *CDH3*, respectively). Apart from their important physiological roles [6,8,9], these proteins have been highly implicated in oncogenesis. Considering epithelial cancers, the loss of E-cadherin expression is considered a prerequisite for tumor cell invasion and dedifferentiation [11,12,13,14] and it frequently parallels the de novo expression of mesenchymal cadherins, such as N-cadherin, a process called “cadherin switch”, which is a marker of the epithelial-to-mesenchymal transition (EMT) molecular process [15,16,17,18]. Nevertheless, evidence from other tumors has raised questions regarding this simplified model of cadherins’ expression in cancer [19,20,21,22]. Tissue-specific expression profiles of cadherins suggest that, similarly to its cancer-related functions, cadherin expression subtypes must be context dependent, particularly when considering the added role brought by interactions between these proteins [23]. Interestingly, current knowledge on the EMT process parallels this concept of dynamic cadherins’ expression, with transitory- and function-dependent regulation of gene expression, resulting in a complex non-binary program, frequently reaching a hybrid phenotype between epithelial and mesenchymal states [24]. Importantly, the presence of various subclones with distinct epithelial, mesenchymal or intermediate/hybrid states within the same tumor has been observed [25], which strongly contributes to tumor heterogeneity.

In the central nervous system (CNS), the role of classical cadherins in gliomas remains to be clarified, as published data remain limited and conflictive [26,27,28,29]. Gliomas are primary brain tumors notorious for their invasiveness, diffuse growth and treatment resistance [30,31]. Glioblastoma (GBM), the most frequent glioma, is a heterogeneous and aggressive tumor, with a median overall survival of 15 months [32,33]. In 2010, transcriptomic analysis of glioblastoma established the presence of a molecular classification into four subtypes: proneural, neural, classical and mesenchymal [34]. Since then, the mesenchymal subtype of glioblastoma has been thoroughly investigated and has been suggested to be associated with an invasive phenotype, increased aggressiveness and robust therapy resistance [35,36]. However, how the mesenchymal transcription subtype aligns with a phenotypic mesenchymal transition, or how the concept of EMT applies to GBM and to other tumors with non-epithelial origin, is still uncertain. In particular, the canonical EMT-related E- to N-cadherin switch is unlikely to be a central process in these neoplastic diseases and the assumption that N-cadherin expression marks for such a mesenchymal phenotype is likely farfetched. A recent review highlights how our current knowledge about cadherin expression in gliomas arises mostly from small sampled series, with significant contradicting results [29]. For instance, N-cadherin expression has been associated with both a protective role, with a decrease in tumor migration, invasiveness and dissemination [27,37,38,39], and with a detrimental role, with a documented association between increased N-cadherin expression and high pathological glioma grade [40,41], the presence of radioresistant glioma stem cells [42], and a significant decrease in patient survival [28,41]. Taken together, the published results collectively point to a broad N-cadherin expression in gliomas, with a likely increase from low-grade to high-grade gliomas [41,43]. E-cadherin expression in gliomas is reduced and suggested to be more frequent in low-grade gliomas than in GBM [44,45]. Interestingly, a recent paper studied, for the first time, P-cadherin expression on gliomas and demonstrated increased expression in high-grade tumors and a correlation with a decreased overall survival in GBM [46]. Moreover, P-cadherin expression was shown to induce an increased tumorigenic behavior, invasive capacity and stem-like characteristics in GBM cell lines [46].

Overall, the conclusions of published results on the biological role of cadherins’ expression in GBM dedifferentiation, dissemination or disease aggressiveness are thus far mostly contradictory and, as a consequence, highly inconclusive. In order to further clarify this topic, we herein present the analysis of cadherins’ expression in a large series of GBM and correlate their expression profiles with clinical, imaging and survival data. In addition, we explored the interplay of cadherins’ expression and how it could improve the understanding of GBM’s unknown pathogenesis.

## 2. Materials and Methods

### 2.1. Patients and Materials

A retrospective series of 313 patients who underwent biopsy or surgical resection for newly diagnosed glioblastomas (GBM), between 2010 and 2017, at the Neurosurgery Department, Centro Hospitalar de Santo António (Porto, Portugal), has been retrieved (Appendix A). Formalin-fixed and paraffin-embedded (FFPE) tumor tissues were histologically analyzed for all primary tumors, as well as for matched local recurrences in 38 patients. Clinical data, treatment decision and patient and survival follow-up data were extracted by a review of patients’ records for all the cases included in the series. Initial MRI was also evaluated and imaging characteristics were documented. This study has been performed in accordance with the national regulative law for the handling of biological specimens from tumor banks, as well as under the international Helsinki declaration. Moreover, it has also been approved by the ethics committee at Centro Hospitalar de Santo António (DEFI 104-17 (093-DEFI/091-CES)). The study design is illustrated in Appendix A.

### 2.2. Neuropathological Evaluation

All cases were reviewed by an experienced neuropathologist, blinded to clinical and demographic details. Neuropathological features were analyzed from hematoxylin and eosin staining of the initial tumor samples. Neuropathological phenotypes were evaluated and the presence of distinct areas of differentiation was documented according to the following features: (1) the glial component, which included agglomerates of fibrillary astrocytes, gemistocytes, multinucleated giant cells or cells with oligodendroglial features; (2) the epithelioid component, which was considered when there was a morphologic epithelial appearance, when large and polygonal shape cells were present, and when there was abundant cytoplasm and a defined cell border; (3) a sarcomatous component was also considered, whenever areas of sarcomatous or mesenchymal differentiation were observed, often comprising chondral or osteochondral elements. The predominant component (>50% of the tumor) was documented as well.

### 2.3. Immunohistochemical Staining

Immunohistochemical (IHC) staining was performed with specific monoclonal antibodies for the classical cadherins, N-, E-, and P-cadherins, in 3 μm sections: E-cadherin (1:100, rabbit, clone 24E19, Cell Signalling, Danvers, MA, USA), P-cadherin (1:50, mouse, clone C56, BD Transduction, Franklin Lakes, NJ, USA) and N-cadherin (1:100, mouse, clone 3B9, ThermoFisher, Waltham, MA, USA). Briefly, dewaxing and epitope recovery (ER) was performed using Dewax and HIER Buffer M (Epredia, Kalamazoo, MI, USA) and all ERs were heat-induced in an Epredia PT Module, according to the standard protocol. The downstream IHC protocol was performed in the Lab Vision^TM^ Autostainer 480S, using the Epredia UltraVision™ Quanto-HRP Detection System. The time for primary antibody incubation was 60 min, with the exception of the anti-P-cadherin antibody, where a 90 min incubation was performed. All reactions were revealed with 3,3′-diaminobenzidine (DAB) chromogen. Tissues were then counterstained with HIGHDEF^®^ hematoxylin (Enzo, Farmingdale, NY, USA), dehydrated and coverslipped using a permanent mounting solution (Entellan^®^). Positive and negative controls were included in every set of reactions for each antibody used. The expression for each cadherin was evaluated according to staining intensity [+: weak (1), ++: moderate (2), +++: strong (3)], staining area [<25% (1), 25–50% (2), >50% (3)] and membrane staining [positive (2) or negative (1)]. All the analyses were blinded for clinical and demographic details. An algorithm was then used with the sum of intensity, extension and membrane positivity values, as defined in Appendix A. Whenever the cadherin expression score was >3, it was considered positive. This scoring system was applied for all cadherins. In tumors comprising various paraffin blocks, a tumor was codified as positive for a given cadherin if positivity was observed in at least one block. In addition to considering isolated positivity for E-, P- and N-cadherin, we further classified the tumors according to the expression of the 3 cadherins in 8 subgroups: (1) isolated N-cadherin expression; (2) isolated E-cadherin expression; (3) isolated P-cadherin expression; (4) E- and N-cadherin co-expression; (5) P- and N-cadherin co-expression; (6) E- and P-cadherin co-expression; (7) positivity for the 3 cadherins (triple-positive); and (8) negativity for the 3 cadherins (triple-negative).

### 2.4. Data Mining from Publicly Available Datasets

RNA expressions from a total of 154 glioblastoma multiform patients (TCGA-GBM) were downloaded from genomic data commons along with the available clinical information on 3 April 2023. Expression data in FPKMs (fragments per kilobase of exon per million mapped fragments) for *CDH1*, *CDH2* and *CDH3* were extracted from the RNA counts file provided by TCGA data. Additional clinical information (MGMT status, methylation class, G-CIMP methylation, IDH1 status, expression subclass and therapy class) was extracted from the original paper [47]. All the analyses performed in this report were carried out in R version 4.1.2. Expression was classified into high and low categories using the median value of expression as the threshold.

### 2.5. Statistics

The chi-square test was used to assess group differences, as appropriate. Overall survival from GBM diagnosis to patient death or last follow-up was estimated using the Kaplan–Meier limit method. The log-rank test was used to assess group differences. A 2-tailed significance level (*p*-value) of 0.05 was applied. Graph Pad Prism version 9.0c software (Graph Pad Software, San Diego, CA, USA) was used for statistical analysis and graphical data presentation.

## 3. Results

### 3.1. Clinical and Neuropathological Characteristics of Glioblastoma Series

Patient characteristics, including clinical baseline parameters and treatment decisions, are detailed in Appendix A. The series of 313 GBM patients include a similar number of male (n = 178; 56.9%) and female (n = 135; 43.1%) individuals, with a mean age at diagnosis of 61,743 ± 10,483 years old. Regarding presentation at diagnosis, the majority of patients had focal deficits (n = 116; 37.1%), followed by headaches (n = 74; 23.6%), behavior changes (n = 65; 20.8%) and seizures (n = 54; 17.3%). The mean for the initial functional status, measured by the Karnofsky Performance Scale (KPS), was 79.33 ± 10.585. Concerning tumor MRI characteristics, we could describe that the mean major axis of the tumors was 7.27 ± 2.09 cm. The majority were unilobar (n = 242; 77.3%), mostly located in the frontal (36.1%) and temporal lobes (30.4%), without ependymal (n = 224; 71.6%) or corpus callosum extension (n = 221; 70.6%). Less than a third of the tumors (n = 76, 24.3%) were multifocal and 49 (15.7%) presented cysts. The majority of patients underwent surgical resection (n = 177, 56.5%), followed by chemotherapy (temozolomide) and radiotherapy, as proposed by Stupp et al. (64.9%) [48]. We documented tumor recurrence or regrowth in 297 patients. Of these, 38 patients underwent surgical resection and 159 patients underwent second-line treatments.

The neuropathologic features that we analyzed are summarized in Table 1. Necrosis and vascular changes were ubiquitous in primary tumors. We identified the presence of inflammation in about a third of the cases (n = 80; 25.6%). A glial component was present in up to 99.7% of the tumors (n = 312); in particular, gemistocytes were identified in 15.7% (n = 49) and an oligodendroglial component in 11.7% (n = 36). A mesenchymal component was identified in 20.1% of the tumors (n = 63), more commonly than the presence of an epithelial component, which was only found in 13.1% (n = 41). Based on the outlined criteria, tumors were divided into predominantly glial (n = 291; 93%), predominantly mesenchymal (n = 15; 4.8%) or predominantly epithelial (n = 7; 2.2%).

### 3.2. Cadherins’ Expression Associated with Clinical, Imaging and Neuropathological GBM Features

We have studied the expression of the classical cadherins, E-, N- and P-cadherin, in the series of 313 GBMs (Figure 1A). In cases where preserved normal brain parenchyma was present, we observed N-cadherin expression in neurons and ganglion cells, as expected; however, no staining was found in the brain parenchyma for either E- or P-cadherin (Figure 1A).

N-cadherin was the most commonly expressed cadherin in up to 81.5% of the tumor samples (n = 255) (Table 1). The majority of these samples had strong N-cadherin staining, since compound scores were higher than five in 49.9% of the cases (n = 156). Moreover, membrane staining was observed in 64.5% of cases (n = 202). In the available matched recurrence cases, virtually all tumors were N-cadherin positive (n = 37; 97.4%), where strong staining was observed in 77% of the cases (n = 30) (Table 1, Figure 1C). Interestingly, N-cadherin was ubiquitously expressed among tumors with different neuropathological components: glial (254 in 312 cases, 81%), mesenchymal (44 in 63 cases, 70%) and epithelial (36 in 41 cases, 88%) (Table 1, Figure 1B). Importantly, N-cadherin expression was not significantly associated with any particular patient or tumor characteristic.

Positivity for E-cadherin was only found in 31% (n = 97) of the cases (Table 1). In particular, E-cadherin staining was more frequent in the cytoplasm of larger cells, with an abundant cytoplasm and defined cell borders, with occasional membrane staining (10.3%) (Figure 1A). Only 9.4% of the cases had compound scores higher than five (n = 31). Upon tumor recurrence, contrary to what was observed for N-cadherin, E-cadherin staining loss was observed in a large number of re-operated patients with initial positivity for E-cadherin (43.6%). E-cadherin positivity dropped to 20.5% (n = 8) in the recurrent cases (Table 1, Figure 1C). Interestingly, E-cadherin positive tumors were more frequently observed in younger patients (46.7% positivity in patients <43 years old, *p* = 0.079), characteristically located in the temporal lobe (48%, *p* = 0.007), with the presence of cysts (*p* = 0.028) and a tendency for cortical involvement (*p* = 0.094). The neuropathological characteristics that significantly correlated with immunostaining for E-cadherin included the presence of papillary structures (*p* < 0.001) and the presence of inflammatory infiltrates (*p* = 0.001). Strikingly, E-cadherin was positive in 95.1% (39 in 41 cases) of the cases with an epithelial component (Table 1). In cases with a predominant epithelial component, E-cadherin was consistently expressed in all cases (7 in 7 cases) (*p* < 0.001). On the other hand, E-cadherin was positive in 26.7% (4 in 15 cases) of the cases with predominant mesenchymal differentiation and in 29.5% (86 in 291 cases) of the cases with predominant glial differentiation (Table 1, Figure 1B).

Immunostaining for P-cadherin was only found in 20.8% of primary GBMs (n = 65) (Table 1). As for E-cadherin, P-cadherin was also found in the cytoplasm of large cells, with low membrane staining (6.1% of the cases) (Figure 1A). In addition, it commonly appeared in tumors containing fusiform cells (Figure 1A). Strikingly, P-cadherin’s positivity rate in recurrence was significantly higher compared to the primary tumors (65.8% vs. 20.8%) (Table 1, Figure 1C). Comparing matched primary and recurrent tumors, P-cadherin gain was the most frequent event, present in 51.3% of the tumor matched pairs. The presence of a mesenchymal component also significantly correlated with positive staining for P-cadherin (*p* < 0.001), which was positive in up to 66% of these cases. It was also positive in 23% of the cases with an epithelial component (Table 1, Figure 1B). Indeed, positivity for P-cadherin was found in 60% of the tumors where the mesenchymal component was predominant (9 in 15 cases), in 42.9% where there was a predominance of the epithelial component (3 in 7 cases), and in 18.2% of predominant glial tumors (53 in 291 cases) (*p* < 0.001). In addition, we identified the presence of lymphocytic infiltrates in about 50% of P-cadherin positive cases (*p* < 0.001).

### 3.3. Cadherins’ Co-Expression Profiles Improve the Identification of GBM Subgroups

We further classified tumors into eight subgroups according to the co-expression of the three studied cadherins. The expression of more than one cadherin was identified in 124 tumors (39.6%), while only 22 tumors expressed all the three cadherins (triple-positive tumors, 7%). Moreover, the immunostaining was negative for N-, E- and P-cadherin in 40 tumors (12.8%, triple-negative tumors). Most commonly, tumors expressed only N-cadherin (n = 137, 43.8%) or co-expressed E- and N-cadherin (n = 67, 21.4%), as shown in Figure 2 and Figure 3A. Interestingly, upon tumor recurrence, the most common profiles were the co-expression of P- and N-cadherin (46.2%) or the isolated expression of N-cadherin (30.8%) (Figure 3A). Cadherins’ expression profiles changed in 69.2% of the cases. Curiously, all E-cadherin positive cases, upon tumor recurrence, showed concomitant expression of P- and N-cadherin (Figure 3A).

In tumors with more than one FFPE block, we evaluated the heterogeneity of each cadherin subgroup. Over 50% of the tumors showed intratumoral cadherin expression heterogeneity, including four distinct subgroups in 1.7% of the tumors. Curiously, N-cadherin expression was heterogeneous in 40% of the tumors and E-cadherin in 28% of the tumors, while P-cadherin was heterogeneous in only 16% of the tumors.

When considering the combined expression of the three cadherins, we further evaluated putative significant correlations with clinical, imaging and neuropathological characteristics. Clinical presentation was associated with cadherin co-expression profiles (Figure 2). In particular, patients with tumors that expressed epithelial cadherins more frequently presented focal deficits (P-cadherin, *p* = 0.024 and co-expression of E- and P-cadherin, *p* = 0.018), while headache was the most common presentation in patients with tumors that showed isolated E-cadherin expression (*p* = 0.015). These expression profiles also correlated with age; specifically, the isolated expression of E-cadherin was more common in younger patients (*p* = 0.015). Tumor location was more common in the frontal lobe for isolated N-cadherin expression tumors (*p* = 0.024) and the temporal lobe for the co-expression of E- and N-cadherin (*p* = 0.022). Other imaging features correlated with cadherin expression subgroups included cyst formation (co-expression E- and N-cadherin, *p* < 0.001, and a tendency for E-cadherin, *p* = 0.057), multifocality (P-cadherin expression, *p* = 0.011, and co-expression P- and N-cadherin, *p* = 0.024), ependymal extension (P-cadherin expression, *p* = 0.030, and E- and P-cadherin co-expression, *p* = 0.036), and corpus callosum extension (tendency for P- and N-cadherin co-expression, *p* = 0.055).

When evaluating for distinct neuropathological phenotypes, we observed that tumors with a predominant sarcomatous/mesenchymal differentiation predominantly exhibited N- and P-cadherin co-expression (44.4%) or were positive for the three cadherins (33.3%) or expressed only P-cadherin (11.1%) or only N-cadherin (11.1%) (*p* < 0.001) (Figure 3B). Predominant epithelial differentiation was observed in tumors showing only E-cadherin expression (28.6%), E- and N-cadherin co-expression (28.6%) and the expression of the three cadherins (triple-positive, 42.9%) (*p* < 0.001) (Figure 3B). Tumors with predominant glial expression most frequently had only N-cadherin expression (46.1%), E- and N-cadherin co-expression (21.5%) or were negative for the three cadherins (13.5%) (Figure 3B).

Based on the phenotypic differentiation results, we further explored cadherin expression differences according to transcription subtypes, particularly the mesenchymal subtype, using the online available TCGA data. Indeed, these data showed that tumors with the mesenchymal transcription subtype were enriched for isolated N-cadherin expression (28%), P- and N-cadherin co-expression (16%), triple positivity (14%), triple negativity (14%) and isolated P-cadherin expression (12%) subgroups. When compared to tumors with other subtypes, P- and N-cadherin co-expression was specifically associated with the presence of a mesenchymal subtype (*p* = 0.019) (Figure 3C).

### 3.4. Cadherins Expression Profiles Define GBM Patients’ Prognosis

As expected, patients’ age (HR = 1.405; *p* = 0.004), functional status (KPS 50; HR = 1.772; *p* = 0.006), surgical decision (biopsy; HR = 2.329; *p* < 0.001) and complimentary treatment (no treatment; HR = 4.73; *p* < 0.001) were validated as prognostic factors in this particular GBM series (Table 2). Thus, survival analysis was used to correlate cadherins’ expression with GBM patients’ prognosis.

In particular, N-cadherin expression identified tumors with improved progression-free survival (median 6 months, log rank test, *p* = 0.033) and overall survival (median 12 months, log rank test, *p* = 0.001) (Figure 4A), while a significant decrease in progression-free survival and overall survival was observed in tumors positive for P-cadherin expression (log rank test, median 3 months and 7 months, *p* = 0.028 and *p* = 0.024, respectively) (Figure 4C). E-cadherin staining identified tumors with a longer time to recurrence (7 months, log rank test, *p* = 0.008) and a better overall survival (14 months, log rank test, *p* = 0.039) (Figure 4B).

Interestingly, the combined expression of cadherins also stratified patients’ prognosis, both progression-free and overall survival (Figure 4D and Table 2). For instance, Kaplan–Meier survival curves showed a survival benefit for E- and N-cadherin co-expression for both progression-free (log rank test, *p* = 0.005, median 7 months) and overall survival (log rank test, 14 months, *p* = 0.005) (Figure 4D). On the contrary, P- and N-cadherin or E- and P-cadherin co-expression showed shorter times to recurrence (median 2 and 3 months, log rank test, *p* = 0.046 and *p* = 0.093, respectively), while E- and P-cadherin co-expression and single P-cadherin expression showed a tendency for worse overall survival (log rank test, 9 months, *p* = 0.078 and 9 months, *p* = 0.055, respectively) (Figure 4D).

Univariate analysis identified three subgroups with worse progression-free survival: E- and P-cadherin co-expression (HR = 2.203, *p* = 0.035), P- and N-cadherin co-expression (HR = 1.91, *p* = 0.008) and triple-negative tumors (HR = 1.712, *p* = 0.021) and four subgroups with worse overall survival: E- and P-cadherin co-expression (HR = 2.728, *p* = 0.020), P-cadherin (HR = 2.611, *p* = 0.012), triple-negative tumors (HR = 1.794, *p* = 0.005) and P- and N- co-expression (HR = 1.619, *p* = 0.036) (Table 2).

Multivariate analysis identified E- and P-cadherin co-expression (HR = 2.96, *p* = 0.020) as a significant independent factor for worse progression-free survival (Table 3). E-cadherin (HR = 4.24, *p* = 0.017), E- and P-cadherin co-expression (HR = 3.35, *p* = 0.021), P-cadherin (HR = 2.33, *p* = 0.036) and triple positivity (EPN) (HR = 2.11, *p* = 0.011) were also found as significant independent factors for worse overall survival (Table 3). In concordance with these results, of the 21 long-term survivors (LTS) (6.7%) within our series, 42.9% had N- and E-cadherin co-expressing tumors, compared to the 19.9% expression in non-LTS (*p* = 0.013); further, 38.1% of the GBMs showed N-cadherin expression, 9.5% were triple-positive and other 9.5% were triple-negative tumors. Strikingly, the tumors from LTS patients did not show isolated expression of epithelial cadherins (E- and P-cadherins) (Figure 3D).

## 4. Discussion

E-cadherin, P-cadherin and N-cadherin expression function and impact in prognosis is well established in various systemic cancers. It is nevertheless important to remember that this consensus is accepted for epithelial tumors and, although E- to N-cadherin switch is frequently used as a marker of EMT in cancers of other origins, the factual involvement of these proteins in processes akin to the EMT in non-epithelial tumors remains to be adequately studied and clarified. Actually, the EMT process recently became accepted as a continuum between these two definable states, often achieving hybrid phenotypes, with increased cell plasticity. In this context, cadherin expression regulation will most likely follow the cell–cell adhesion requirements of the different EMT states.

Given the knowledge gap in cadherin expression in glial tumors, our main goal was to characterize the expression of the most studied classical cadherins (E-, N- and P-cadherins) in a large and representative series of GBMs. To do so, and taking into account GBM intratumoral heterogeneity, we marked the expression of these proteins in all available paraffin blocks of 313 patients, in a total of 727 samples. Our main findings were the following: (1) N-cadherin is the most frequent cadherin expressed in GBM, present in over 80% of the cases, followed by E-cadherin in about a third of the tumors and P-cadherin in 20%; (2) expression of epithelial cadherins (E- and P-cadherin) determines a distinct imaging and neuropathological profile in GBM; and (3) expression of epithelial cadherins has a significant detrimental impact on GBM patient’s prognosis. In addition to showing distinct associations with clinical and neuropathological characteristics, the distinct behavior of E- and P-cadherins’ expression upon tumor recurrence is notable. There was a clear P-cadherin dominance and a loss of E-cadherin expression in recurrent lesions compared with primary tumors. Survival analysis further supported P-cadherin as a marker of tumor aggressiveness, as P-cadherin-expressing tumors were significantly correlated with a worse patient prognosis. On the contrary, E-cadherin expression was observed in patients with a significant increase in progression-free and overall survival.

These results, although corroborated by robust morphological correlations, only considered the expression of each isolated cadherin and may thus represent a simplified version of an intricately regulated cellular mechanism between them. In fact, the mechanistic reciprocities between the expression of distinct cadherins have been already suggested, although poorly explored in tumor pathogenesis. Examples include the co-expression of R- and E-cadherin that suppresses the progression and metastasis in salivary adenoid cystic carcinomas [49], the combined expression of cadherin-6 and cadherin-11 associated with lymph node metastasis and poor prognosis in oral squamous cell carcinoma [50], and the co-expression of E- and P-cadherin which is associated with poor prognosis in breast cancer [23]. In particular, further studies of the mechanisms involving E- and P- co-expression led to the proposal of P-cadherin as a marker of an intermediate EMT phenotype [51].

Hence, we expanded our analysis and evaluated tumor characteristics considering the simultaneous expression of the three classical cadherins. This allowed us to not only identify distinct tumor subgroups considering clinico-imaging and neuropathological features but, more importantly, to confirm the importance of cadherin’s interplay in the biological outcome of GBM. For example, the isolated expression of N-cadherin was able to identify the most common GBM profiles (up to 43.8% of the cases), which are the ones commonly found in the frontal lobe, with a typical glial component and few inflammatory infiltrates. Importantly, when N-cadherin expression is combined with E- or P-cadherin, divergent tumor subtypes are observed: E- and N-cadherin co-expressing tumors are frequently located in the temporal lobe, with cortical involvement and cysts, and frequently exhibit an epithelial component, while GBMs positive for P- and N-cadherin frequently display a mesenchymal component, with frequent immune infiltrates. Finally, survival analysis highlights tumors with E- and N-cadherin co-expression as a subgroup with significantly improved progression-free and overall survival, while the P- and N-cadherin subgroups have shorter times to recurrence and worse overall survival. It is also noteworthy that, in our data, P-cadherin’s increase in recurrence is always accompanied by N-cadherin expression, while co-expression of E-cadherin and N-cadherin was not observed in any recurrent case. Tumors without N-cadherin expression, but with the expression of epithelial cadherins, commonly exhibit distinctive neuropathological and MRI patterns and collectively are subgroups with a worse prognosis.

Taken together, these results suggest that although N-cadherin is the dominant cadherin in GBM, its function can be strongly modulated by the expression of epithelial cadherins. When associated with E-cadherin, it frequently leads to an epithelial-differentiated tumor with a distinct clinical profile and a survival benefit. However, when combined with P-cadherin expression, it leads to tumors with a predominant mesenchymal differentiation, with reduced time for recurrence and survival. In addition, this is the most frequent subgroup upon tumor recurrence. These results led us to hypothesize that the mesenchymal state in glioblastoma is most likely reflected by the combined expression of P- and N-cadherins. In fact, these results were further supported by data leveraged from the TCGA data, showing a significant association between the mesenchymal transcriptional profile and P- and N-cadherin co-expression.

P-cadherin positive status, in our analysis, clearly marks for the worse prognosis subgroups. In particular, isolated P-cadherin or E- and P-cadherin co-expression were found as independent predictors for worse overall survival, while also displaying an imaging profile suggestive of multifocality and propensity to dissemination. This is in concordance with recent reports which associated P-cadherin expression with GBM aggressiveness, through an increase in cell invasion and migration, as well as a decrease in patient survival [46]. In addition, we curiously observe that these tumors often display both mesenchymal and epithelial differentiation, but not a homogeneous differentiation, denoting perhaps the presence of heterogeneous cancer cell populations with increased phenotypic plasticity. Thus, our results suggest that neither E-cadherin, nor N-cadherin, expression is able to counteract the disadvantageous biology brought by P-cadherin expression. An apparent profile with increased cancer cell plasticity and markers for enhanced infiltrative potential seems to be denoted by the expression of epithelial cadherins, either P-cadherin expression or E- and P-cadherin co-expression, which we interpret in the context of the possible acquisition of a hybrid EMT state.

The role of E-cadherin, also an epithelial cadherin, with infrequent expression in GBM, is less clear. Our results show a potential dichotomy in the role of this protein. On the one hand, when it was co-expressed with N-cadherin, a subgroup with distinct neuropathological and MRI features and improved survival was identified. On the other hand, tumors with isolated expression of E-cadherin seemed to portend a worse prognosis, although they exhibited a similar neuropathological and imaging profile to E- and N-cadherin-expressing tumors. Published reports on epithelioid GBMs describe this neuropathological and imaging profile and typically identify tumors with an aggressive course [52,53]. It is thus possible that, as described for other tumors, the expression of other concomitant cadherins, and hence activation of different cellular programs, and not expression of E-cadherin per se, will define overall tumor characteristics and aggressiveness.

Finally, the biology of triple-positive and triple-negative tumors is equivocal. Triple-negative tumors do not relate to specific patient characteristics or prognosis. Triple-positive tumors present all three morphological components, although are significantly associated with the predominance of a mesenchymal component. They also are a frequent subgroup in tumor recurrence. These results point to a closer similarity to the P- and N-subgroup. In accordance with these results, we propose a reformulation of the rigid concept of E- to N-cadherin switch as a marker of EMT activation and tumor aggressiveness in GBM. In fact, we hypothesize that tumor characteristics are more likely dictated by the interplay between the expression of different cadherins and considering the expression of a single cadherin may contribute to equivocal results. N-cadherin tumors are the most frequent in our analysis and reflect the average GBM. Our results propose P-cadherin and N-cadherin interplay as a contributor for the mesenchymal-like state in GBM. The presence of an intermediate, more plastic phenotype, with increased tumor aggressiveness, is most likely reflected by the presence of epithelial cadherins, P-cadherin or E- and P-cadherin co-expression. Lastly, GBM is not an epithelial tumor and is therefore less obvious if an epithelial differentiation would present a protective effect. If so, then such putative protective epithelial state in GBM is clearly reflected by an E- and N-cadherin interaction. Further studies on the mechanisms underlying cadherin interplay are warranted.

In addition, upon validation in other series, our results propose cadherin expression subgroups as a reflection of tumor biology and EMT states. Therefore, they become putative biomarkers to guide clinical decision. These results will become particularly interesting with additional characterization of EMT pathways in GBM and identification of treatment targets.

## 5. Conclusions

In this study, we document the expression of N-, E- and P-cadherin in GBM. Our findings establish that cadherin co-expression is associated with consistent clinical, imaging and neuropathological differences, ultimately correlating with patient prognosis. Additionally, we challenge the notion that an E- to N-cadherin switch is a definitive marker of an EMT-like program in GBM. Instead, we propose that cadherin co-expression subgroups serve as indicators of distinct differentiation stages within GBM.

## Figures and Tables

**Figure 1 cancers-16-02298-f001:**
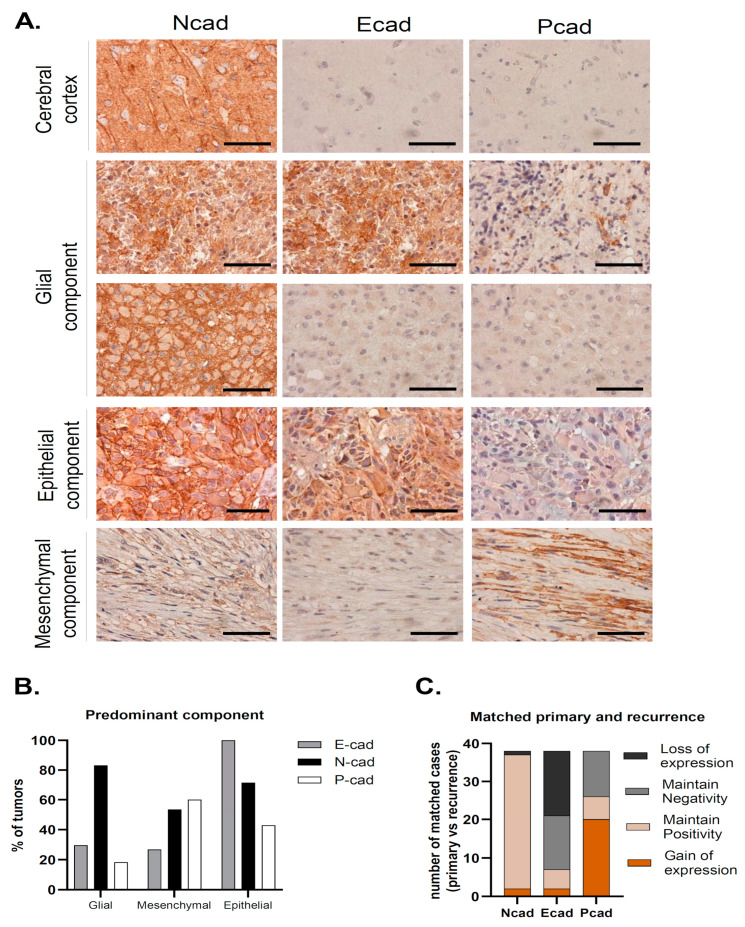
N-, E- and P-cadherin expression in Glioblastoma. (**A**) Representative cases of predominant glial, epithelial and mesenchymal component tumors. In tumors with a predominant glial component, N-cadherin positivity was observed in the majority of the cases and was frequently strong, with membrane staining. In some cases, E-cadherin positivity was also observed. Tumors with a predominant epithelial component stained for E-cadherin in all cases, which was frequently accompanied by N-cadherin staining, as observed in this case. Finally, tumors with a predominant mesenchymal component most commonly stained for both N-cadherin and P-cadherin (scale bar corresponds to 50 μm). (**B**) N-cadherin expression was high in all tumor subtypes. In tumors with a predominant mesenchymal component, P-cadherin and N-cadherin expression was observed in the vast majority of cases. Strikingly, all cases with a predominant epithelial component were positive for E-cadherin. (**C**) Upon tumor recurrence, 97.4% of the tumors were positive for N-cadherin (maintain positive + gain of positivity). Gain of P-cadherin expression was the most common event which was observed in over half of the cases (51.3%). This leads to an important increase in P-cadherin positivity in recurrence when compared to the primary tumor (20.8% vs. 65.8%). On the contrary, E-cadherin positivity is lower in recurrence (31% vs. 20.8%), due to E-cadherin loss of staining in matched primary-recurrent cases (43.6%).

**Figure 2 cancers-16-02298-f002:**
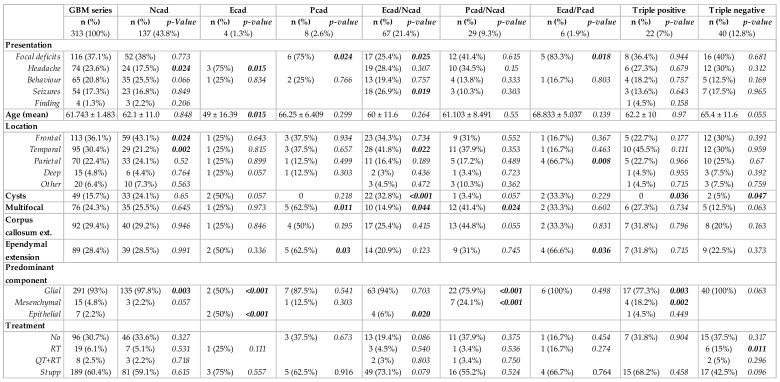
Cadherins’ co-expression profiles in glioblastomas and association with clinical, imaging and neuropathological characteristics.

**Figure 3 cancers-16-02298-f003:**
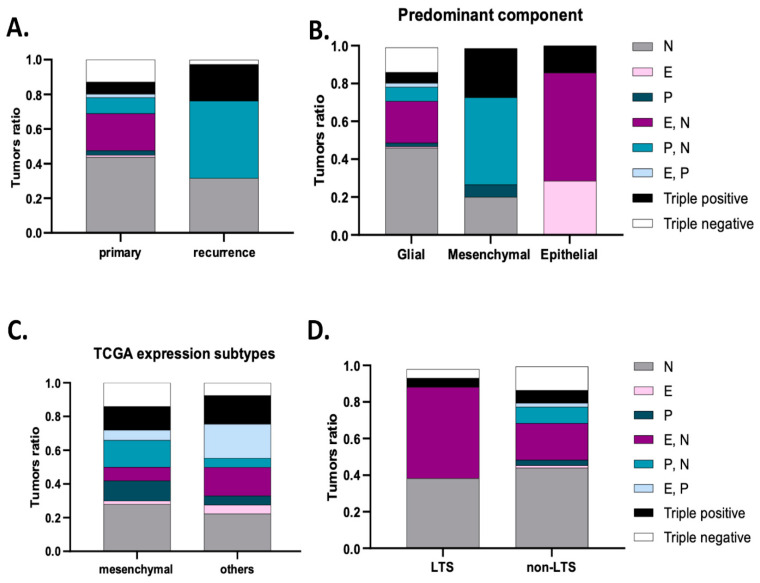
Cadherins’ expression subgroups in glioblastoma. (**A**) Compared to primary tumors, the most frequent subgroups in recurrent tumors are P- and N-cadherin co-expression (46.2%) and N-cadherin expression (30.8%). (**B**) Tumors with a predominant epithelial component expressed E-cadherin, E- N-cadherin or triple-positive subgroups. For mesenchymal tumors, the subgroups expressed were P- N-cadherin, triple-positive, P-cadherin and N-cadherin. (**C**) Analysis of data from the TCGA data confirmed a main expression of P- and N-cadherin, N-cadherin, triple-positive and N-cadherin subgroups in the mesenchymal transcriptional subtype. P- and N- co-expression significantly correlated with the mesenchymal subtype (*p* = 0.033). (**D**) Tumors from long-term survival patients more frequently exhibited E- and N-cadherin co-expression, N-cadherin, triple-positive or triple-negative subgroups (*p* = 0.013).

**Figure 4 cancers-16-02298-f004:**
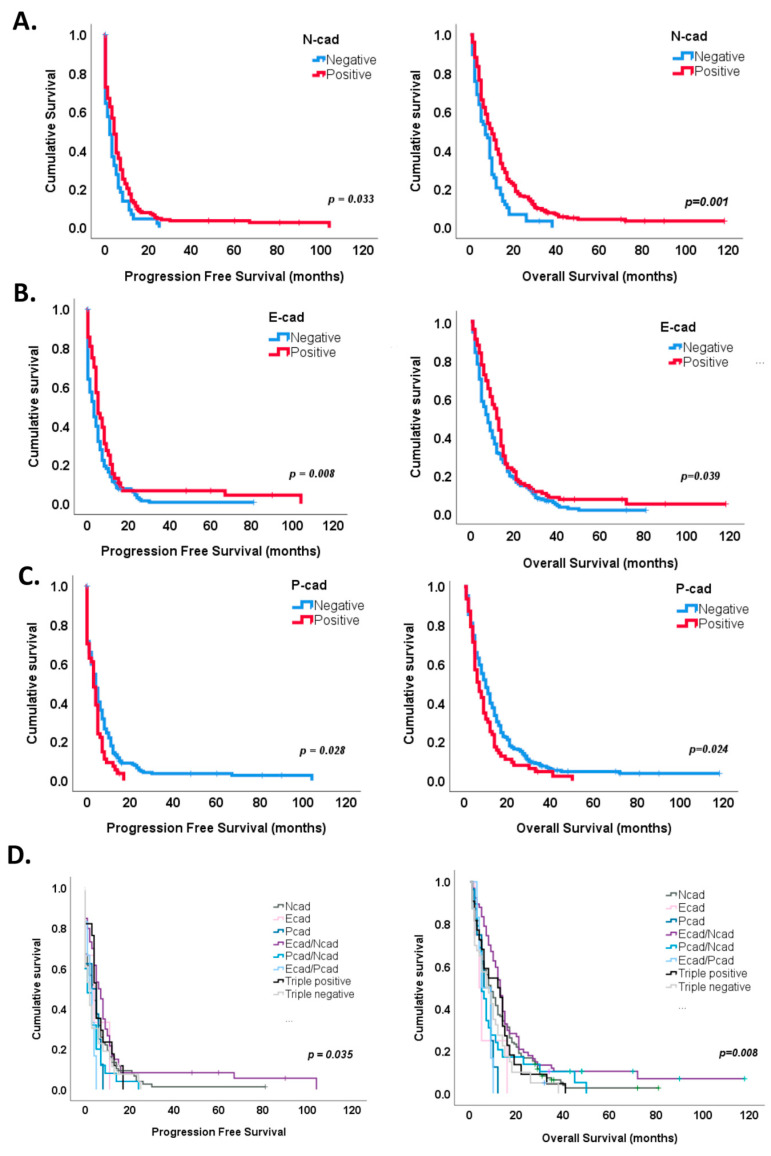
Survival analysis according to cadherins’ expression. (**A**) N-cadherin expression in GBM showed a significant improvement in both progression-free (*p* = 0.033) and overall survival (*p* = 0.001). (**B**) A significant survival benefit was also observed in tumors with E-cadherin expression (*p* = 0.008 for progression-free survival and *p* = 0.039 for overall survival. (**C**) On the contrary, P-cadherin expression significantly marked tumors with a shorter time for recurrence (*p* = 0.028) and survival (*p* = 0.024). (**D**) Kaplan–Meier curves identified a survival benefit for E- and N-cadherin co-expression subgroup (*p* = 0.005 for both PFS and OS). P-, N-cadherin co-expression (*p* = 0.046) subgroup significantly associated with a worse progression-free survival, while E-P-cadherin co-expression (*p* = 0.078) and P-cadherin (*p* = 0.055) subgroups showed a tendency for worse overall survival.

**Table 1 cancers-16-02298-t001:** Individual cadherins expression in GBM and association with neuropathological features.

	GBM Series	N-Cadherin Positive		E-Cadherin Positive		P-Cadherin Positive	
	n (%)	n (%)	*p*-Value	n (%)	*p*-Value	n (%)	*p*-Value
	313 (100%)	255 (81.5%)		97 (31%)		65 (20.8%)	
**Papillary structures**	17 (5.4%)	15 (5.9%)	0.46	17 (17.5%)	**<0.001**	6 (9.2%)	0.126
**Whorls**	8 (2.6%)	5 (2%)	0.162	4 (4.1%)	0.239	6 (9.2%)	**<0.001**
**Glomeruloid vessels**	47 (15%)	39 (15.3%)	0.773	20 (20.6%)	0.063	9 (13.8%)	0.767
**Myxoid stroma**	18 (5.7%)	16 (6.3%)	0.404	9 (9.3%)	0.072	4 (6.2%)	0.875
**Inflammation**	80 (25.6%)	69 (27%)	0.502	39 (40.2%)	**0.001**	33 (50.8%)	**<0.001**
**Calcifications**	10 (3.2%)	10 (3.9%)	0.125	3 (3.1%)	0.945	0	0.1
**Oligodendroglial component**	36 (11.7%)	32 (12.5%)	0.643	9 (9.3%)	0.299	0	**0.014**
**Gemistocytes**	49 (15.7%)	49 (19.5%)	**<0.001**	13 (13.4%)	0.424	5 (7.8%)	0.058
**Glial component**	312 (99.7%)	254 (99.6%)	0.633	96 (99%)	0.135	65 (100%)	0.608
**Mesenchymal component**	63 (20.1%)	44 (17.3%)	0.75	25 (25.8%)	**0.005**	43 (66.2%)	**<0.001**
**Epithelial component**	41 (13.1%)	36 (14.1%)	0.263	39 (40.2%)	**<0.001**	15 (23%)	**0.007**
**Predominant component**							
*Glial*	291 (93%)	242 (94.9%)	0.981	86 (88.7%)	**<0.001**	53 (81.5%)	**0.001**
*Mesenchymal*	15 (4.8%)	8 (3.1%)	0.561	4 (4.1%)	0.376	9 (13.8%)	**<0.001**
*Epithelial*	7 (2.2%)	5 (2%)	0.489	7 (7.2%)	**<0.001**	3 (4.6%)	0.145
**Tumor recurrence**	38	37 (97.4%)		8 (20.5%)		25 (65.8%)	

**Table 2 cancers-16-02298-t002:** Univariate analysis of cadherins’ expression profile in glioblastomas and association with patient’s prognosis.

		Progression-Free Survival				Overall Survival			
		HR	95.0% CI		*p*-Value	HR	95.0% CI		*p*-Value
			Inferior	Superior			Inferior	Superior	
**Surgery**	**Surgery**	1				1			
	**Biopsy**	2.120	1.626	2.763	**<0.001**	2.329	1.842	2.945	**<0.001**
**Treatment**	**Yes**	1				1			
	**No**	13.244	8.492	20.665	**<0.001**	4.730	3.547	6.308	**<0.001**
**KPS**	**100**	1				1			
	**50**	0.505	0.056	4.550	0.543	1.772	1.183	2.654	**0.006**
	**60**	0.580	0.078	4.318	0.595	1.411	0.866	2.298	0.167
	**70**	0.587	0.080	4.295	0.600	2.575	1.223	5.421	**0.013**
	**80**	0.430	0.059	3.105	0.403	2.596	1.035	6.508	**0.042**
	**90**	0.391	0.054	2.826	0.352	1.599	1.022	2.503	**0.040**
**Age**	**< 65 yo**	1				1			
	**> 65 yo**	1.258	0.974	1.625	0.079	1.400	1.113	1.762	**0.004**
**Ncad**	**Negative**	1				1			
	**Positive**	0.725	0.522	1.007	0.055	0.627	0.468	0.840	**0.002**
**Ecad**	**Negative**	1				1			
	**Positive**	0.716	0.546	0.938	**0.016**	0.779	0.608	0.998	**0.048**
**Pcad**	**Negative**	1				1			
	**Positive**	1.367	1.003	1.863	**0.048**	1.364	1.029	1.806	**0.031**
**Cadherins co-expression**	**Ecad/Ncad**	1				1			
	**Ncad**	1.367	0.984	1.900	0.063	1.316	0.969	1.787	0.079
	**Ecad**	1.507	0.471	4.826	0.490	2.550	0.923	7.041	0.071
	**Pcad**	1.864	0.884	3.933	0.102	2.611	1.239	5.502	**0.012**
	**Pcad/Ncad**	1.910	1.186	3.076	**0.008**	1.619	1.033	2.536	**0.036**
	**Ecad/Pcad**	2.203	1.068	5.868	**0.035**	2.728	1.169	6.366	**0.020**
	**Triple positive**	1.194	0.683	2.087	0.533	1.427	0.875	2.328	0.154
	**Triple negative**	1.712	1.084	2.704	**0.021**	1.794	1.196	2.690	**0.005**

**Table 3 cancers-16-02298-t003:** Multivariate analysis of cadherins’ expression profile in glioblastomas and association with patient’s prognosis.

		Progression-Free Survival				Overall Survival			
		HR	95.0% CI		*p*-Value	HR	95.0% CI		*p*-Value
			Inferior	Superior			Inferior	Superior	
**Mesenchymal Component**	**yes**	1.000				1.000			
	**no**	1.060	0.670	1.660	0.811	0.810	0.530	1.250	0.345
**Epithelial Component**	**yes**	1.000				1.000			
	**no**	0.930	0.580	1.470	0.743	0.720	0.450	1.140	0.160
**Cadherins co-expression**	**Ecad/N-cad**	1.000				1.000			
	**Ncad**	0.990	0.690	1.440	0.979	1.300	0.900	1.870	0.162
	**Ecad**	2.200	0.650	7.430	0.205	4.240	1.290	13.930	**0.017**
	**Pcad**	1.330	0.600	2.980	0.482	2.330	1.050	5.130	**0.036**
	**Pcad/Ncad**	1.340	0.770	2.360	0.303	1.440	0.850	2.460	0.178
	**Ecad/Pcad**	2.960	1.180	7.400	**0.020**	3.350	1.330	8.440	**0.010**
	**Triple positive**	1.190	0.640	2.210	0.577	2.110	1.190	3.760	**0.011**
	**Triple negative**	1.090	0.660	1.820	0.728	1.490	0.910	2.450	0.113
**Surgery**	**Surgery**	1.000				1.000			
	**Biopsy**	1.760	1.290	2.390	**<0.001**	1.550	1.170	2.060	**0.003**
**Treatment**	**yes**	1.000				1.000			
	**no**	15.000	9.020	24.930	**<0.001**	4.570	3.300	6.330	**<0.0001**
**KPS**	**100**	1.000				1.000			
	**50**	0.150	0.020	1.390	0.094	0.700	0.080	6.170	0.752
	**60**	0.250	0.030	1.900	0.179	2.400	0.320	18.040	0.396
	**70**	0.240	0.030	1.790	0.162	2.190	0.300	16.220	0.442
	**80**	0.210	0.030	1.550	0.125	1.520	0.210	11.150	0.681
	**90**	0.200	0.030	1.510	0.118	1.310	0.180	9.660	0.790
**Age**	**<65 yo**	1.000				1.000			
	**>65 yo**	1.210	0.920	1.600	0.176	1.270	0.980	1.660	0.072

## Data Availability

All data generated or analyzed during this study are included in this published article and its Appendix A.

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
