# Peer review of "Cadherin Expression Profiles Define Glioblastoma Differentiation and Patient Prognosis"

_cancers, 2024, doi:10.3390/cancers16132298_

Round 1

Reviewer 1 Report

Comments and Suggestions for Authors

The authors evaluate the expression patterns of Cadherins 1,2, and 3 in Glioblastoma (GBM), and identify prognostic correlations with important clinicopathological variates in this malignancy. The authors conduct histological analysis of resected/biopsied patient GBMs including matched primary and recurrent tumors in multiple cases, and validate their results using GBM TCGA RNAseq datasets. The experimental design is robust, and correlations with most relevant patient variables are adequately addressed. The manuscript builds on previous work by the same group and others, and adds new insights into an expanding field looking at the role of EMT-like programs in driving Glioblastoma.

Comments:

1.While the authors conduct cadherin expression correlation analysis across various clinicopathological factors (Table 2), the data will be strengthened further by including treatment decision as a variable. Within the context of this study, this information is crucial in understanding if treatment modality (particularly in the case of recurrent tumors) is correlated with specific cadherin expression patterns (eg: E-cadherin loss and P-cadherin gain in recurrent tumors as demonstrated in the data)

2. When correlating cadherin expression with patient prognosis (Figure 3, Tables 3-4), it is suggested the authors add information about extent of resection (EOR) for each patient if available. This is important since EOR could determine PFS and overall survival independent of cadherin expression patterns. 

3. While the data suggests a correlation between P-cadherin positivity and worse prognosis (independent, or co-expressed with N/E cadherin), can the authors comment on why cadherin triple-negativity is also correlated with worse OS?

4. The correlations with clinical presentation (Table 2) are unclear (headaches, focal deficits, and the undefined measures of behavior and finding).  It is suggested the authors use specific clinically relevant presentation measures (e.g tumor size) to draw correlations with cadherin expression and provide more concrete justifications (e.g imaging data) as to why cadherin expression and neuropathological differences reflect different EMT states in GBM (as written in line 41-42 of the main text) 

5. Can authors comment on the therapeutic implications of this study in the discussion section?

Comments on the Quality of English Language

Minor comments: 

1.Change commas to periods throughout manuscript and tables (percentages, etc).

2.Correct spelling to “Temozolomide” in line 198 of the main text.

Author Response

1. While the authors conduct cadherin expression correlation analysis across various clinicopathological factors (Table 2), the data will be strengthened further by including treatment decision as a variable. Within the context of this study, this information is crucial in understanding if treatment modality (particularly in the case of recurrent tumors) is correlated with specific cadherin expression patterns (eg: E-cadherin loss and P-cadherin gain in recurrent tumors as demonstrated in the data)

The authors thank the reviewer’s comment and agree that treatment information is crucial to interpret the results obtained. Thus, we have now included the information regarding tumor treatment modalities in Table 2. Actually, we were also curious to understand if a given treatment modality would predominantly induce a specific post-treatment/recurrent cadherin profile; however, all evaluated recurrent tumors received adjuvant treatment according to the Stupp protocol.

 2. When correlating cadherins expression with patient prognosis (Figure 3, Tables 3-4), it is suggested that the authors add information about extent of resection (EOR) for each patient if available. This is important, since EOR could determine PFS and overall survival independent of cadherin expression patterns. 

 The authors acknowledge the reviewer’s comment and agree that the extent of resection would be a relevant variable to include in the survival analysis. Unfortunately, extent of resection as determined by post-operative MRI only became routine in 2016; therefore, we lack this relevant analysis for the majority of our patients. We tried to overcome this lack of information including the variable surgery vs. biopsy in the multivariate analysis (Table 3).

3. While the data suggests a correlation between P-cadherin positivity and worse prognosis (independent, or co-expressed with N/E cadherin), can the authors comment on why cadherin triple-negativity is also correlated with worse OS?

The authors thank the reviewer’s comment. Indeed, the biology underlying triple negative tumors is, regarding the important lack of knowledge in cadherins in GBM, difficult to speculate. Nevertheless, the correlation between triple negative tumors and worse prognosis was not significant in the multi-variate analysis and thus, perhaps, denotes other associated characteristics apart from lack of cadherins expression.

4. The correlations with clinical presentation (Table 2) are unclear (headaches, focal deficits, and the undefined measures of behavior and finding). It is suggested the authors use specific clinically relevant presentation measures (e.g tumor size) to draw correlations with cadherin expression and provide more concrete justifications (e.g imaging data) as to why cadherin expression and neuropathological differences reflect different EMT states in GBM (as written in line 41-42 of the main text).

The authors appreciate the reviewer’s comment and the possibility to clarify our results and manuscript. We evaluated clinical presentation as there is some evidence that it correlates to tumor biology and patient prognosis, for instance considering presentation with seizures vs focal deficits (Pallud et al 2024). We evaluated initial MRI with various characteristics, including the VASARI features, which includes tumor major and minor axis. We have presented in the results that were significant or identified as relevant. Moreover, our hypothesis that cadherin expression reflects different EMT states in GBM is based primarily on the consistent observation of particular neuropathological features among different cadherin expression subgroups. Then, we curiously observed that, in addition to these neuropathological profiles, their expression was still associated with unique MRI features (such as cysts, multifocality or ependymal extension) and some clinical characteristics, most notably the difference in age observed for the E-cad subgroup. We have reviewed our manuscript and clarified this information. 

 5. Can authors comment on the therapeutic implications of this study in the discussion section?

 The authors acknowledge the reviewer’s comment and thank for the opportunity to improve the manuscript. We have included and highlighted in our discussion the following sentence: Upon validation in other series, our results propose cadherin expression profiles as a reflection of tumor biology and EMT states. Therefore, they became putative biomarkers to guide clinical decisions. These results will become particularly interesting with additional characterization of EMT pathways in GBM and identification of therapy targets.”

Reviewer 2 Report

Comments and Suggestions for Authors

This is a very important research work that contributes to the field of neuroncolgy. This work evaluated the presence of basic cadherins in glioblastoma samples and in the final survival of the patients. In general, the manuscript was well written and the study design and the analysis data are adequate.  I suggest to the authors to add a figure to summarize the design of the study.

Author Response

This is a very important research work that contributes to the field of neuroncology. This work evaluated the presence of basic cadherins in glioblastoma samples and in the final survival of the patients. In general, the manuscript was well written and the study design and the analysis data are adequate. I suggest to the authors to add a figure to summarize the design of the study.

The authors thank the reviewer’s comment and appreciation of the study. We have improved our manuscript including 2 new supplementary figures: one referring to patient selection (Supplementary Figure 2) and the other clarifying the design of the study (Supplementary Figure 3).

Reviewer 3 Report

Comments and Suggestions for Authors

In the current work, Dr. Noronha and her colleagues performed a comprehensive analysis of the expression profiles of N-, E-, and P-cadherins in histologic material from 313 GMB patients and analyzed the association of their expression with a wide range of characteristics, including tumor node location, patient age, neuropathologic features, associated disorders (headache, focal deficits, etc.), predominant components, progression-free survival, and overall survival. The results obtained were further verified by transcriptome analysis of data deposited in the TCGA database. As a result of this study, the authors obtained unique new results indicating (a) the need to extend the standard E-to-N-cadherin switch accompanying mesenchymal transition of tumor cells, as other types of cadherins (particularly P-cadherin in GBM) may also play a key role in determining cell phenotype; (b) high correlation of P-cadherin overexpression (alone or in combination with other cadherins) with worse prognosis in GBM patients; (c) no correlation between high expression of epithelial cadherins and high survival of GBM patients (patients with co-expression of E- and N-cadherins had the best prognosis). Considering the good level of this study, well-prepared text of the article and illustrative material, I recommend this paper for publication in Cancer. Authors should pay attention to minor comments only:

Line 140 - please add a space between 3 and um

Tables 1-3 - use periods instead of commas to separate fractions.

I thank the authors for their work and wish them continued success! Based on this work, the association of P-cadherin with glioblastoma aggressiveness identified by Noronha et al. will be integrated into the upcoming studies of our research group.

Author Response

In the current work, Dr. Noronha and her colleagues performed a comprehensive analysis of the expression profiles of N-, E-, and P-cadherins in histologic material from 313 GMB patients and analyzed the association of their expression with a wide range of characteristics, including tumor node location, patient age, neuropathologic features, associated disorders (headache, focal deficits, etc.), predominant components, progression-free survival, and overall survival. The results obtained were further verified by transcriptome analysis of data deposited in the TCGA database. As a result of this study, the authors obtained unique new results indicating (a) the need to extend the standard E-to-N-cadherin switch accompanying mesenchymal transition of tumor cells, as other types of cadherins (particularly P-cadherin in GBM) may also play a key role in determining cell phenotype; (b) high correlation of P-cadherin overexpression (alone or in combination with other cadherins) with worse prognosis in GBM patients; (c) no correlation between high expression of epithelial cadherins and high survival of GBM patients (patients with co-expression of E- and N-cadherins had the best prognosis). Considering the good level of this study, well-prepared text of the article and illustrative material, I recommend this paper for publication in Cancer. Authors should pay attention to minor comments only:

Line 140 - please add a space between 3 and um

Tables 1-3 - use periods instead of commas to separate fractions.

I thank the authors for their work and wish them continued success! Based on this work, the association of P-cadherin with glioblastoma aggressiveness identified by Noronha et al. will be integrated into the upcoming studies of our research group.

The authors thank the reviewer’s kind words and the manuscript was modified according with the suggested alterations.